# Obesity-Related Cross-Talk between Prostate Cancer and Peripheral Fat: Potential Role of Exosomes

**DOI:** 10.3390/cancers14205077

**Published:** 2022-10-17

**Authors:** Shangzhi Feng, Kecheng Lou, Cong Luo, Junrong Zou, Xiaofeng Zou, Guoxi Zhang

**Affiliations:** 1The First Clinical College, Gannan Medical University, Ganzhou 341000, China; 2Department of Urology, The First Affiliated Hospital of Gannan Medical University, Ganzhou 341000, China; 3Institute of Urology, The First Affiliated Hospital of Ganna Medical University, Ganzhou 341000, China; 4Jiangxi Engineering Technology Research Center of Calculi Prevention, Ganzhou 341000, China

**Keywords:** adipose tissue around prostate cancer, adipose exosomes, tumor microenvironment, obesity, prostate cancer

## Abstract

**Simple Summary:**

Obesity is involved in many aspects of prostate cancer progression as a risk factor for prostate cancer, especially in the process of biochemical recurrence in the prostate. Approximately 27–53% of prostate cancer patients can develop biochemical recurrence after radical prostatectomy, which poses difficulties in the clinical management of prostate cancer, and this is closely related to the release of exosomes from adipose tissue in the obese state. In this review, we summarize the crosstalk between prostate cancer peripheral adiposity and prostate cancer and discuss the potential role of exosomes in this process and the prospects for the use of adipose exosomes. Exosomes play an important role in the crosstalk between the two this may be a new basis to explain obesity as a biochemical recurrence after prostate cancer surgery and a potential avenue for future prostate therapy.

**Abstract:**

The molecular mechanisms of obesity-induced cancer progression have been extensively explored because of the significant increase in obesity and obesity-related diseases worldwide. Studies have shown that obesity is associated with certain features of prostate cancer. In particular, bioactive factors released from periprostatic adipose tissues mediate the bidirectional communication between periprostatic adipose tissue and prostate cancer. Moreover, recent studies have shown that extracellular vesicles have a role in the relationship between tumor peripheral adipose tissue and cancer progression. Therefore, it is necessary to investigate the feedback mechanisms between prostate cancer and periglandular adipose and the role of exosomes as mediators of signal exchange to understand obesity as a risk factor for prostate cancer. This review summarizes the two-way communication between prostate cancer and periglandular adipose and discusses the potential role of exosomes as a cross-talk and the prospect of using adipose tissue as a means to obtain exosomes in vitro. Therefore, this review may provide new directions for the treatment of obesity to suppress prostate cancer.

## 1. Introduction

Prostate cancer (PCa) is the most common cancer among men in developed countries. It is also one of the cancers with significantly increased incidence in developing countries [1,2]. The reasons for the increasing incidence of PCa in Asian countries year by year cannot be explained solely by an increase in the level of screening and diagnosis based on the implementation of PSA. This increased incidence of PCa also largely reflects the rise in risk factors associated with PCa. Obesity is associated with a more aggressive tumor, poorer treatment outcomes, and a higher risk of mortality for PCa patients [3,4]. Several clinical studies have indicated that a higher visceral adiposity is associated with higher grade or aggressiveness of PCa [5,6]. Various mechanistic studies of obesity-mediated cancer progression have further shown that obesity influenced the aggressiveness of PCa due to increased systemic inflammation, hyperinsulinemia, altered adipokine profile, and increased lipid function [7,8]. Although there are very few adipocytes in the prostate [9], clinical and in vitro laboratory studies have evaluated the relationship and potential mechanisms between periprostatic carcinoma adipose (PPAT) and PCa progression [10,11,12,13] (Table 1). 

Obesity is associated with the incidence of different cancers, poor treatment outcomes, and high cancer-related mortality rates [29]. Although individual studies have found that the incidence of PCa, lung, colorectal, and ovarian cancer is inversely associated with patient BMI [30]. Beyond that, a significant association between obesity and PCa progression has been established, especially in relation to more malignant and biochemical recurrence (biochemical recurrence defined as patients without postoperative endocrine therapy and radiotherapy, with two consecutive prostate-specific antigen (PSA) ≥ 0.2 μg/L during follow-up) of PCa [31,32,33]. Meta-analyses have also shown that obesity is related to more advanced prostate cancer [4,34]. Cells from tissues surrounding or adjacent to prostate cancer, such as stromal cells, endothelial cells, immune cells, and adipose stem cells, can form a tumor microenvironment that promotes prostate cancer survival [35]. This also indicates that peri-Pca adipose tissue and its tumor mesenchymal cells can promote Pca growth through various mechanisms, including the release of growth factors, inflammatory signaling activators, and the release of fatty acids to provide energy for Pca. Besides the specific regulation of these biological factors, exosomes secreted by peri-Pca adipocytes may also play a key role in Pca growth. For example, changes in the amount and status of white adipose tissue (WAT) and brown adipose tissue (BAT) caused by obesity affect the amount and cargo of exosomes released into circulation. The unhealthy expansion of adipose tissue in the obese state leads to local hypoxia, which induces chronic inflammation. Adipocyte dysfunction then induces changes in exosomes [36], thus mediating PCa progression through various molecular mechanisms [37].

As a new mode of cellular communication, extracellular vesicles contain cellular molecules with various physiological roles. They can be divided into two categories based on their size: vesicles of 50–1000 nm formed via germination and exosomes of 40–160 nm from the somatic membrane [38]. Exosomes have many sources, including all living cells containing RNA, bioactive lipids, and proteins [39], with various functions related to biological processes. Exosomes participate in pathological and physiological regulatory processes by transporting these substances. Besides the pro-carcinogenic role of tumor mesenchymal adipocyte exosomes, peritumor adipose tissue may also communicate this process through exosomes. Exosomes may play a key role in the cross-talk between PCa and peri-tumor adipose tissue. However, studies have not comprehensively evaluated the PPAT phenotype and its detailed cross-talk with PCa. Furthermore, the role of exosomes in the association between PCa and peri-cancerous fat is unclear. It is also unclear whether removal of this fat and inhibition of exosome release is beneficial for patients and whether obesity exacerbates the ability of exosomes to mediate this process.

## 2. Cross-Talk between PCa and PPAT

PPAT is the adipose tissue surrounding the surface of the prostate (Figure 1). PPAT varies in thickness and structure at different locations in the prostate [40]. However, PCa cells communicate with adipose tissue cells in the same way (defined as extraperitoneal extension of the tumor) after PCa invasion. This process is usually associated with a significantly poorer prognosis for PCa [41,42]. Adipocytes within the omentum promote the initial homing of ovarian cancer cells to the omentum and provide fatty acids to cancer cells, thus promoting rapid tumor growth by secreting adipokines [43]. PPAT, which is external to the cancerous tissue, also plays a role in PCa progression. Studies of PPAT thickness, expression of inflammatory factors, and the involvement of obesity in prostate cancer progression have shown that changes in local adipose tissue in the prostate body can influence the behavior of PCa [13]. In addition, PCa induces pro-tumorigenesis in peripheral adipose tissue. For example, preadipocytes triggered by PCa cell-culture medium undergo tumor-like transformations, including genetic variability, epithelial–mesenchymal transition (EMT), and tumor-like lesion formation in vivo [44]. This suggests that the continuous positive feedback process between PCa and PPAT accelerates the continuous deterioration of PCa.

### 2.1. PPAT Promotes Survival and Progression of Prostate Cancer

Like the obesity epidemic, the understanding of adipocytes and adipose tissue is expanding. Major advances in the last decade have provided new insights into the role of adipose tissue in normal physiology and obesity-related complications. As a result, adipocyte biology studies have focused on the global metabolic disease pandemic [45]. Adipocytes can secrete various effectors, including exosomes, miRNAs, lipids, and inflammatory cytokines, which play an important role in paracrine and endocrine influencing local and systemic metabolic responses. Adipose tissue acts as an endocrine organ or embedded energy store to actively promote tumor growth and metastasis by secreting extracellular matrix components, such as adipokines, pro-inflammatory cytokines, and pro-angiogenic factors [21,46,47,48]. However, obesity exacerbates this process. A study reported that obesity is associated with a poorer PCa prognosis [49]. Obesity may lead to the development of high-grade prostate cancer by increasing aromatase activity and regulating the secretion of various cytokines such as adipokines, vascular endothelial growth factors, and prostaglandins [50,51]. However, it is unclear how the interaction and single effect of multiple factors influence the tandem process between obesity and PCa.

The main biologic factors released by adipocytes (adipokines and inflammatory factors) are recognized biological mediators of cross-talk between PPAT and PCa. Secretion of multiple biologic factors by adipocytes was associated with a significant increase in the proliferation, migration, and invasive capacity of PCa cells [21,46,52]. In one study, adipokine CCL7 overexpression was detected in prostate specimens collected from obese patients after puncture and radical PCa surgery. Adipokine CCL7 was positively correlated with PCa pathological malignancy than normal punctured prostate tissue, suggesting that PPAT can enhance the malignancy of PCa cells [52]. PPAT contains MMP (matrix metalloproteinase: plays an important role in many cellular behaviors) with higher activity than abdominal visceral adipose tissue [21]. Activated MMP can degrade extracellular matrix proteins and thus promote the invasion of cancer cells into surrounding tissues [53]. In vivo experiments also revealed that mature epididymal adipocytes from rats or humans with PCa accelerated the growth and differentiation of normal rat or human prostate epithelial cells. These are also accompanied by enhanced expression of VEGF (vascular endothelial growth factor) and PdGF (platelet-derived growth factor) in prostate epithelial cells and activation of the PI3K pathway in PC3 cells [54,55]. Adipose tissue in obese patients is always in a state of chronic inflammation, characterized by overexpression of inflammatory factors secreted by adipose tissue [56], including IL-6 (interleukin-6: cytokines involved in the inflammatory response), IL-8 (interleukin-8: cytokines involved in the inflammatory response), MCP-1 (monocyte chemoattractant protein 1), and TNF-α (tumor necrosis factor α), possibly due to excessive infiltration of immune cells, such as macrophages in obese tissues [57]. However, further studies should assess whether the degree of inflammation in PPAT is correlated with obesity or metabolic syndrome. Several in vivo and in vitro experiments have shown that these inflammatory factors are closely associated with the progression of PCa [58,59,60]. Another study also showed that the degree of inflammation in PPAT was associated with adipocyte size, insulin and triglyceride circulating levels, and high levels of PCa [61]. Furthermore, the appearance of cachexia in advanced tumor stages may be due to the metabolic changes in adipocytes caused by cancer cells, which leads to lipolytic pathway activation, making adipocytes a key energy source for cancer cells. This situation also applies to PPAT with advanced PCa. For example, the expression of lipolytic enzymes and CGI-58 was more pronounced in late PCa [62]. FTIR microspectroscopy (an optically based technique that can measure the transitions in vibrational modes of the functional groups of bio-molecular constituents within cells) also showed that lipid translocation occurs between adipocytes and PCa cells [63]. NMR spectroscopy (a technique applying nuclear magnetic resonance phenomena for the determination of molecular structure) showed that patients with high-grade PCa have higher rates of monounsaturated/saturated fatty acids than those in low-grade patients [64]. Fatty acids serve as an important energy source for cancer cells, and studying how PPAT affects the uptake and utilization of fatty acids by PCa and the detailed mechanism of action of these fatty acids on PCa may improve the understanding of the mutual cross-linking between obesity and PCa. PPAT can also interfere with the expression of hormones that promote PCa progression. PPAT, as a source of local extra-gonadal androgens, promotes the growth and metastasis of PCa. PPAT also expresses aromatase, which converts androgens to estrogens [65] and plays a crucial role in the pathogenesis and progression of PCa. For example, estrogens can activate wild-type and mutant androgen receptors, limiting ADT treatment [66]. In addition, obesity may exacerbate this process by altering the metabolic and endocrine characteristics of multiple adipose tissues, thereby increasing the release of multiple biological factors, as well as the mobilization of free fatty acids [52,67,68]. Obesity also increases the rate of preadipocyte migration in WAT, leading to obesity-induced PCa progression [69,70]. PPAT is more active in metabolism and secretion in obese men than in lean men. For example, PPAT in obese men has higher MMP-9 activity, which leads to better induction of PC3 cell lineage and endothelial cell proliferation [71]. PPAT from obese men has a higher expression of chemokine CXCL1 than that of lean men [72], which may lead to PCa susceptibility to bone metastasis [73]. 

### 2.2. PCa Affects the State and Production of PPAT

Cancer-associated adipocytes (CAA) are defined as the result of interactions between adipocytes and tumor cells, leading to the remodeling of adipocytes into a more poorly differentiated state of adipocytes, consistent with the finding that PPAT is richer in precursor adipocytes than other visceral adipose tissue [12]. This precursor cell phenotype is associated with more aggressive tumors, including PCa [52,74]. CAA is associated with the exacerbation of some of the malignant features of cancer cells, which ultimately constitute a positive feedback loop with cancer cells in the malignant progression of tumors [21,75]. For example, PCa–conditioned medium (CM) more affects preadipocytes than non-malignant PCa cells. Preadipocytes disturbed by PCa–CM participate in several biological processes of PCa progression by undergoing PCa-like tumor-like transformations [44]. Meanwhile, human PCa cells injected into the right side of thymus-free nude mice can be recruited to transplant preadipocytes on the other side. This preadipocyte migration enhances tumor growth and angiogenesis [76]. Some researchers have suggested that PCa cells can also differentiate into adipocyte-like cells and exert PPAT-like pro-cancer effects [77]. Furthermore, CAA undergoes considerable morphological and functional alterations during cancer progression. CAA undergoes dilapidation and acquires a fibroblast-like phenotype in cancer cells, especially at the front of tumor invasion [78]. This phenotypic change is often accompanied by increased secretion of adipocyte differentiation marker proteins (adiponectin, leptin and fatty acid binding proteins, and intestinal proteins) as well as the pro-inflammatory cytokines IL-6 and PAI-1 (plasminogen activator inhibitor-1). This creates a tumor microenvironment that promotes a shift in tumor cell phenotype (increased aggressiveness) [78]. For example, the expression of inflammatory factors (IL-6, IL-8, G-CSF) is significantly higher in PPAT than in serum. IL-6 expression is 375 times higher in PPAT than in serum and significantly correlates with pathological disease grade [13]. This suggests the possibility that PCa and PPAT are interrelated and is potential evidence for the evolution of PPAT into tumor-associated fibroblasts [79]. 

Visceral and subcutaneous adipose tissue are also associated with more aggressive pathological features of PCa. The interaction of PCa with adipose metabolism and testosterone and the effects of 5a reductase inhibitors on prostate and peripheral adipose suggest that a cross-talk may exist between periprostatic adipose tissue and tumors [80]. PCa also regulates the production of leptin via fat-resident regulatory T cells and several leptin receptors, which are associated with the quality of periglandular adipose tissue [81,82]. In vitro experiments of differentiated adipocytes and tumor cells without direct contact have also confirmed that paracrine cytokines from tumor cells induce lipolysis in adipocytes and promote the release of free fatty acids [83]. This also alters the fatty acid composition of PPAT. For example, a study compared the fatty acid composition of PPAT from 12 PCa patients and 11 patients with BPH (benign prostatic hyperplasia) confirmed high levels of palmitic acid and dihomo-γ-linolenic acid and low levels of arachidonic acid in PPAT [84]. Similarly, magnetic resonance spectroscopy showed that PPAT from patients with higher levels of PCa contained a higher ratio of monounsaturated/saturated fatty acids in the fatty acids [16,64]. Therefore, studying the specific processes by which PCa interferes with the release and composition of PPAT fatty acids may give potential avenues for intervention and treatment of PCa. In conclusion, these results suggest that PCa can regulate the state of PPAT and the release of biomolecules to accelerate PCa development in various ways.

## 3. Exosomes as a Mediator of Positive Feedback between Adipose Tissue and Cancer

Recent studies related to adipocytes and extracellular vesicles (EVs) of cancer origin have provided new understanding of the interactions between adipocytes and tumors. Adipose exosomes play a crucial role in cancer progression by transporting fatty acids, biooxidative enzymes, protein degrading enzymes, metabolites, and multiple non-coding RNAs into cancer cells [85,86,87,88] (Table 2). Tumor exosomes can also transport miRNAs, circRNAs, adipokines, and inflammatory factors to adipocytes, thus influencing adipose tissue differentiation and substance release. This can make the adipose further evolve into tumor-associated adipocytes, thereby providing an environment for cancer survival and progression (Table 3). Therefore, the exosomes may act as mediators for PPAT-PCa cross-talk.

### 3.1. Tumor Exosomes Mediate the Alteration of Adipose Tissue and Release of Biomolecules

Exosomes contain mRNA, miRNA, lncRNA, circRNA, DNA, and bioactive cellular metabolites. As a result, exosomes are involved in cellular communication and biological regulation by transporting these substances to the target receptor cells [110]. The miRNA profile of tumor-derived exosomes is similar to that of the tumor cells of origin. Dysregulated exosomal miRNAs can promote cancer migration and proliferation [111]. Tumor-derived exosomes deliver specific miRNAs (miRNA-144, miRNA-126, and miRNA-155) from breast cancer cells to peripheral adipocytes to transform resident cells. For example, exosomal miRNA-144 acts as a mediator between tumor cells and adipocytes, promoting the beige/brown differentiation of adipocytes [99,112]. Adipocytes in melanoma patients secrete more specific exosomes (containing more proteins related to fatty acid oxidation (FAO)), which induce metabolic reprogramming of tumor cells to favor FAO and promote cancer cell aggressiveness [85]. Similarly, exosomes from pancreatic cancer cells can also promote lipolysis by activating p38 and extracellular signaling regulation (ERK1/2) to phosphorylate hormone-sensitive lipases. These catabolized lipids provide energy for the survival of nearby tumor cells [109,113]. In addition, the hepatocellular carcinoma cell line HepG2 secretes exosomes containing specific proteins and activated phosphokinases to act on the NF-κB signaling pathway in adipocytes, thereby releasing adipose exosomes that promote hepatocellular carcinoma growth and angiogenesis and recruit more macrophages [114]. Cancer cell exosomes can also transport HOTAIR (HOX transcript antisense RNA) to endothelial cells, thus increasing VEGFA expression to stimulate angiogenesis, explaining why most cancer-cell-derived exosomes can promote angiogenesis [115]. Released exosomes from gastric cancer cells deliver ciRS-133 to preadipocytes, thus regulating preadipocyte differentiation to brown adipocytes by activating the PR structural domain of PRDM16 [114]. In conclusion, cancer cells differentiate adipocytes into CAA via exosomes, providing a potential material basis for their own survival and progression.

### 3.2. Adipose Exosomes Induce the Development of Multiple Tumors

Currently, studies on exosomes have mainly focused on cancer-cell-derived exosomes, and few studies have focused on CAA exosomes. In fact, the exosomes released from adipocytes cross-talked by tumor cells have obvious structural and functional changes [85]. For example, melanoma-associated adipocyte exosomes contain proteins involved in FAO and induce metabolic reprogramming in tumor cells. The number of secreted exosomes and their effect on tumor aggressiveness are further amplified by the dual effect of obesity and cancer, which exacerbates the symbiotic relationship between adipocytes and cancer cells [85]. For example, MMP3 is highly expressed in adipocyte-derived exosomes in obese lung cancer patients, which is usually transferred to lung cancer cells. As a result, MMP3 promotes the activity of MMP9 in lung cancer cells, thus mediating cancer cell invasion in vitro and in vivo [87]. The specific miRNAs are upregulated in CAA exosomes, inducing survival and progression of many cancer cells. For example, miR-21 is significantly upregulated in triple-negative breast and colorectal-cancer-associated adipose exosomes. Exosomes transport miR-21 from adipose tissue to cancer cells, inhibit apoptosis and promote chemoresistance by binding to apoptosis protease-activating factor-1 (APAF1) [89]. Similarly, miR-210 in exosomes released from adipose stem cells promotes endothelial cell proliferation, invasion, and migration, thus mediating tumor angiogenesis by targeting RUNX3 [116]. The miR-27a-3p in adipocyte-secreted exosomes targets ICOS, an immune-microenvironment-related gene that is significantly upregulated in obese lung adenocarcinoma (LUAD) patients, to promote antitumor immunity in LUAD [96]. In addition, exosomes released from adipose tissue in obese mice contain pro-inflammatory proteins. These exosomes can produce potential stimulatory effects in local and peripheral adipose tissue sites, thus mediating the obesity-associated inflammatory response and insulin resistance and promoting the progression of several cancers [117]. Therefore, CAA exosomes play a crucial role in tumor progression.

### 3.3. Exosomes Mediate Cross-Talk between PPAT and PCa 

Adipose tissue distributed in the periphery of the gland does not act directly on cancer cells by releasing multiple cytokines or by creating a hypoxic environment as adipocytes in the interstitial stroma of the gland do. Studies have reported that IL-6 levels are significantly higher in PCa peripheral adipose tissue (PPAT) than in serum (100-fold). Furthermore, IL-6 levels in peripheral blood are not correlated with IL-6 levels in PPAT. Notably, IL-6 levels in PPAT are correlated with higher Gleason scores, while serum IL-6 levels are not correlated with higher Gleason scores [13]. This suggests that PPAT may not affect the level of inflammatory factors in the body and thus the progression of cancer by directly releasing inflammatory factors. Since the prostate is often separated from the surrounding PPAT by a fibromuscular capsule in the physiological state and the surrounding adipose tissue is more distant than the interstitial adipocytes of the tumor, tight cellular junctions and paracrine secretion as well as the long-distance blood transport of tumor cytokines may not be the main ways to affect it [40,118]. So, in what way does PCa mediate this process? It is certain that a more efficient mediator of crosstalk exists between PPAT and PCa. Exploring this mediator and interfering with it may be a new avenue for PCa therapy. Unlike normal stem cells, adipose stem cells cultured with PC cell-conditioned medium (CM) can form prostate-like tumor lesions in vivo and replicate invasive PCa cells in targeted tissues. This may be due to PCa cell-derived exosomes transporting oncogenic-factor-related transcripts (H-ras and K-ras), multiple miRNAs (miR-125b, miR-130b, and miR-155), and the Ras superfamily of GTPases (Rab1a, Rab1b, and Rab11a) [44]. Also, miR-145-containing exosomes released from adipose stem cells exhibit growth-inhibiting properties of PCa cells [119]. Although this contradicts the ability of PCa cell CM of inducing proliferation of PC3 cell lines and endothelial cells, it suggests that bidirectional exosomes can be released from adipose tissue disturbed by PCa cells. However, the role of adipose exosomes in PPAT with PCa is unclear. Regardless, only a few studies have reported on PCa and adipose exosomes. A systematic evaluation and meta-analysis of 86,490 patients with PCa found an association between obesity and biochemical recurrence after radical prostatectomy [120]. Obesity status of “overweight” and “obese” PCa patients is a key risk factor for biochemical recurrence after radical PCa treatment [121]. Therefore, it is crucial to investigate the interaction between PPAT and PCa, especially the role of exosomes, to provide a basis for explaining the susceptibility to biochemical recurrence after radical PCa resection in obese patients.

## 4. Obesity Alters the Number and Structure of Exosomes Involved in the Cross-Talk between Adiposity and Cancer

### 4.1. Obesity Affects the Number and Status of Adipose Exosomes

Adipocytes in obese individuals rapidly grow into mature adipocytes due to the infiltration of inflammatory factors and the construction of a hypoxic environment affecting the various stages of adipocyte differentiation [122,123,124]. Mature adipocytes secrete different populations of extracellular vesicles (EVs), including small extracellular vesicles (sEVs) and large extracellular vesicles (lEVs). The obvious intrinsic proteins of lEVs are caveolin-1, flotillin-2, and β-actin, which are involved in the shedding of microvesicles. sEVs contain endosomal sorting complex Alix, CD9, CD63, and CD81, and have high cholesterol levels [125]. The differential expression of these exosomal surface proteins suggests that they can be used in clinics [126]. During adipose immaturity, adipocytes in the undifferentiated state secrete increased levels of exosomes. These exosomes contain higher levels of arachidonic acid and adipogenic markers (PPARγ and PREF1). Interestingly, adiponectin expression in exosomes was evident at day 15 of adipose precursor cell differentiation. The uptake of these high adiponectin-expressing exosomes by tumor cells accelerates the rapid growth of various tumors, including breast cancer [43]. In vivo studies have shown that injection of adipose-derived exosomes into mice induces insulin resistance, promotes macrophage polarization processes, and stimulates the secretion of pro-inflammatory cytokines, thus promoting the release of adipose exosomes [117]. Therefore, exosome contents depend on the stage of adipocyte differentiation. Obesity can mediate exosome alterations, thus inducing tumor growth by affecting the differentiation potential of adipocytes.

### 4.2. Obesity Alters the Cargo and Function of Adipose Exosomes

Obesity can directly or indirectly induce cancer progression by affecting adipocyte-derived exosomal cargo. Dysfunctional adipocytes in obese subjects impair the assembly and classification of biological components in exosomes. Obesity also alters the metabolic state, thus affecting the payload and function of adipocyte exosomes. For example, a study characterized exosomes released from adipose tissue from seven obese patients (age: 12–17.5 years, BMI: 33–50 kg/m^2^) and five lean patients (age: 11–19 years, BMI: 22–25 kg/m^2^) and found that 55 miRNAs were differentially expressed in obese vs. lean visceral adipose donors (*p* < 0.05; fold change ≥ |1.2|). The obesity-derived exosomes can also downregulate ACVR2B and regulate the TGF-β1/Wnt/β signaling pathway in A549 cells. Exosomes released from insulin-resistant adipocytes through exposure to pro-inflammatory cytokines (highly expressed in the obese state) or adipose tissue isolated from adult subjects with T2D upregulates genes related to migration, invasiveness, and EMT. Such exosomes exhibit pro-cancer stem-cell-like cell formation and pro-invasive and pro-metastatic behaviors in breast cancer cell models compared with insulin-sensitive adipocytes or adipocytes isolated from adipose tissue of non-diabetic subjects [127]. Studies using a palmitate-induced hypertrophic 3T3-L1 adipocyte model found increased levels of miR-802-5p in the secreted exosomes of these adipocytes. In turn, miR-802-5p downregulates heat shock protein 60 (HSP60), upregulates CCAAT/enhancer binding protein (C/EBP) homologs, and enhances oxidative stress and phosphorylation of c Jun NH(2) terminal kinase (JNK) and insulin receptor substrate 1 (IRS1), leading to the development of insulin resistance [128]. The insulin-resistance-induced diabetes and exosomes with highly expressed miR-802-5p in serum are associated with tumor progression [129]. Obesity accelerates the enlargement of the fat pad, leading to hypoxia within the adipose tissue. Hypoxia upregulates adipocyte exosomal proteins associated with metabolic processes. Studies using the 3T3-L1 adipocyte model [130] showed that exosomal proteins related to lipid synthesis (acetyl-CoA carboxylase, glucose-6-phosphate dehydrogenase, and fatty acid synthase) are upregulated under hypoxic conditions (three to four times higher than under normoxic conditions) [130]. Indeed, studies in obese patients have confirmed the effect of obesity on adipocyte-derived exosomal cargo. For example, clinical studies have found that subcutaneous adipocyte-derived exosomes from obese patients contain higher levels of proteins associated with FAO [85]. In addition, exosomes released from large adipocytes can be transferred to small adipocytes, thereby stimulating adipogenesis and hypertrophy of small adipocytes [131]. These exosomes may be transported into the tumor mesenchyme through the bloodstream, inducing maturation and differentiation of cancer-associated adipocytes and thus providing sufficient energy and material for cancer progression.

### 4.3. Adipose Exosomes Mediate PCa Progression in Obese Patients 

Current research on the causes of obesity-mediated cancer progression focuses on soluble factors, such as increased circulating levels of insulin and other growth factors, altered inflammatory status, and the release of inflammatory molecules and fatty acids from dyslipidemia [8,132]. However, obesity also increases the migration rate of preadipocytes in WAT, thereby mediating PCa progression [133]. In the case of PPAT, obesity also alters the gene expression profile of PPAT, promoting proliferation and immune escape of cancer cells. This creates a microenvironment conducive to PCa progression [134]. Obese men have higher levels of angiogenic capacity in PPAT than lean individuals [71]. Several studies have also indicated that adipocytes can directly construct the tumor microenvironment, forming an adipose tissue containing multiple cells, including immune cells, fibroblasts, preadipose stem cell populations, and mature adipocytes. These cells are associated with the formation and development of various cancers [135,136]. Furthermore, increasing evidence has shown that adipocyte EVs have a key role in tumor progression and tumor-cell–adipocyte interactions in obese patients [137,138]. An experiment investigating the effect of 3T3-L1-derived EVs on the invasiveness of PC3 and DU145 cells found that EVs released from adipocytes stimulate the proliferation of PCa cells and significantly enhance their migratory and invasive abilities. This could be because adipose EVs can trigger a rapid glycolytic process, thus increasing glucose uptake, lactate release, and ATP synthesis in PCa cells [37]. EVs also causes AKT activation and HIF-1α stabilization, commonly associated with overproliferation, metastasis, and chemoresistance in PCa [139,140,141,142]. This recent study suggests that exosomes released in the obese state may be important in the onset and progression of PCa. Taken together, these studies bring new insights into the impact of under-studied adipose deposits, especially PPAT, and their released EVs on PCa (Figure 2).

## 5. Conclusions

The biological processes in adipose tissue, which are cancer-associated, are constantly disturbed by cancer, either directly or indirectly, and this in turn accelerates the further spread of cancer cells in some possible ways [50,51,84]. Especially for the peripheral tissues of cancerous tissues, this symbiosis suggests that cancer may not only communicate with cells in the vicinity of the tumor tissue mesenchyme but may also have some crosstalk with tissues relatively distant from the cancerous tissue, such as peripheral adipose. Therefore, it is important to explore the mechanism of this mutual cross-talk to confirm the existence of this relationship and explain obesity as a cancer risk factor. Exosomes are biological vesicles secreted by cells and can mediate cancer progression by transporting various oncogenic molecules [143]. Exosomes, as biological vesicles secreted by cells with substance transport, signaling, and low antigenic and easy circulation characteristics [144], can mediate cancer progression by transporting various oncogenic molecules [145,146]. Also, tumor-derived exosomes can be transported to adipose tissue to participate in adipocyte remodeling [85,146]. These results suggest that exosomes play a key role in the mutual cross-talk between peripheral adiposity and cancer. Obesity exacerbates this process and can directly alter the release of multiple cytokines from adipocytes, thus influencing cancer development and progression. Obesity can also indirectly promote tumor survival and progression by inducing related diseases [133]. Obesity also exacerbates the positive feedback between cancer and adipose tissue by interfering with the production and cargo of adipose exosomes and thus remotely controlling the cross-talk between adiposity and cancer [37]. Therefore, the effects of obesity and adipose exosomes on PCa and PPAT during their mutual crosstalk are bound to be the focus of future studies.

## 6. Discussion

Obesity is a risk factor for many malignancies, including PCa, colon, breast, liver, and pancreatic cancers. The induction of cytokine release, vascular microenvironment formation, and tumor mesenchymal cell remodeling are all common pathways by which obesity mediates cancer progression. Indeed, the complexity of cancer progression suggests that tumor cells can act not only directly on themselves to maintain their own survival but also indirectly by altering the tumor microenvironment including the release of specific cytokines or vesicles to maintain tumor cell survival or progression. In addition, there appears to be a more powerful disturbing effect of tumor cells on peritumor tissues. For example, tumor tissue can not only affect interstitial cells but can also further induce cancer development by potentially interfering with the state of the tissue surrounding the cancer [37,50,147]. However, there are no robust studies on how tumor cells interfere with surrounding tissues and how surrounding tissues mediate the progression of cancer. Similarly, there is still a lack of sufficient evidence on how peri-cancerous fat is disturbed by cancer and how peri-cancerous fat further contributes to cancer progression.

Whether the presence of PPAT as one of the possible factors of PCa progression may influence the further spread of the lesion leading to multiple recurrences of the cancer as well as maintaining high carcinogenicity and carcinogenicity in the body and thus affecting the use of chemotherapeutic drugs, etc. is uncertain. This leads one to consider the need for simultaneous excision of periglandular adipose tissue during radical surgery for PCa. PPAT is more distant than tumor mesenchymal adipocytes. Therefore, they may not interact as mesenchymal cells interact, such as tight junctions and paracrine secretion. Unlike adipose tissues in other parts of the body, inflammatory factors are highly expressed in PPAT [13]. This suggests the existence of a new mediator that is mediating the mutual crosstalk between PCa and PPAT. Here, we propose that exosomes may be an important way to mediate this process. For example, several studies have shown that exosomes released from PCa cells can mediate stem cell remodeling of adipose tissue [145] and adipose stem-cell-derived exosomes mediate the inhibition of PCa growth via PCa stromal cells [119]. In addition, interfering with the production of adipose exosomes may be an effective way to halt the progression of PCa and to inhibit the crosstalk between PPAT and PCa. The drugs that inhibit vesicle production and release and affect the expression of exosome surface membrane proteins can also reduce exosome production. However, these drugs still have deadly side effects, and more research is needed to confirm their exact value in oncology treatment. Many cancer cells, including PCa can release both pro- and anti-cancer bipartite exosomes, and adipocytes can also release exosomes that activate inflammatory responses in the tumor microenvironment and thus inhibit cancer proliferation and spread [148,149,150]. Therefore, the extraction of potential oncogenic exosomes from adipose exosomes may promote tumor targeting therapy. However, isolating and identifying these specific exosomes is difficult, thus limiting the exosome research work. Adipose exosome research has widely employed the method of interfering with the expression of specific oncogenic molecules in exosomes. Drugs interfering with the source cells or exosome delivery disruption tools can effectively inhibit PCa progression.

Exosome therapies overcome the challenges associated with cellular therapies, including immune rejection, tumorigenic controversy, tissue-specific antigen targeting on the membrane surface, low immunogenicity, and non-invasive passage through the tissue barrier. In addition, adipose tissue may be potentially more promising for use than other tissue exosomes, as follows. (1). Tumor-derived exosomes can activate macrophages in adipose tissue. More monocytes/macrophages may be recruited to adipose tissue after the activation of resident macrophages, further enhancing the inflammatory response. This inflammatory environment further promotes the proliferation of adipocytes and increases the number of associated exosomes. The increased adipocyte size caused by obesity may result in the release of more exosomes from adipose tissue. The combined effect is more efficient for exosome acquisition. (2). The ease of access to adipose tissue and the simplicity of ethical review (Figure 3). Furthermore, liposuction, the most common plastic surgery procedure in the 20th century, can obtain large amounts of human-derived adipose tissue [151]. This access to large amounts of adipose tissue offers the possibility of bulk acquisition and use of human-derived exosomes. (3). Current research on exosomes focuses on MSC exosomes, especially on the regenerative repair functions of MSC exosomes, such as improvement of recalcitrant hair loss, post-traumatic skin healing, spinal cord injury repair, and osteoarthritis [119,152,153]. A prospective, faceted, randomized placebo-controlled trial confirmed that adipose MSC exosomes can be used in skin brightening. The preparations containing ASC exosomes significantly reduce melanin levels than placebo [154]. These studies indicate that the use of MSC exosomes is very promising, and the production of MSC exosomes is bound to become a major concern, and adipose tissue, as the most important source of known MSCs, is bound to become a priority for future sources of large-scale production of stem cell exosomes.

## Figures and Tables

**Figure 1 cancers-14-05077-f001:**
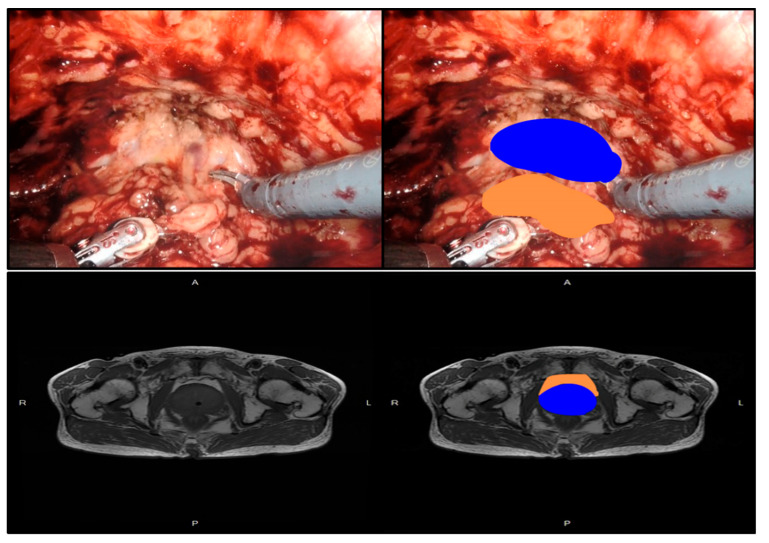
PPAT in robot-assisted radical prostate cancer surgery. The orange color in the image represents PPAT and the blue color represents prostate cancer.

**Figure 2 cancers-14-05077-f002:**
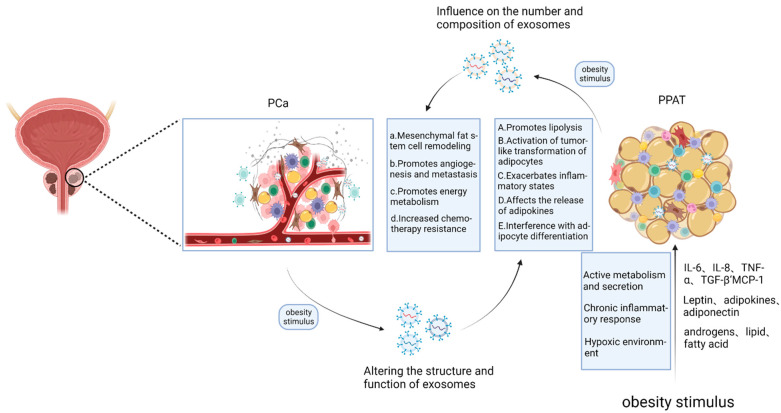
Potential role of exosomes in the intercommunication between PCa and PPAT in the obese state.

**Figure 3 cancers-14-05077-f003:**
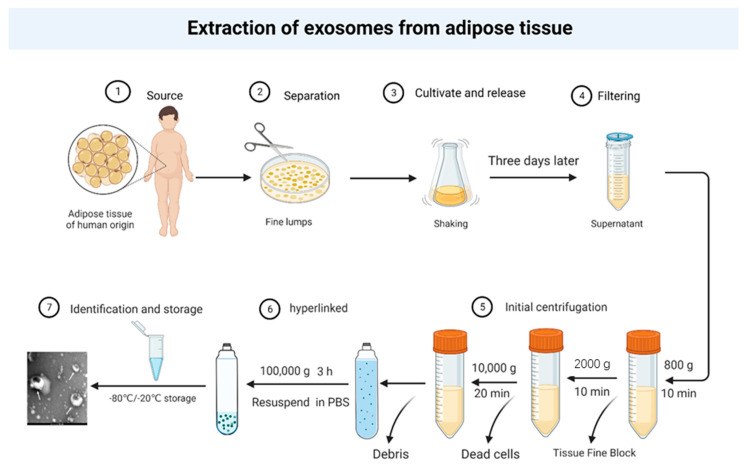
A simple extraction protocol for adipose exosomes.

**Table 1 cancers-14-05077-t001:** Cross-talk between PPAT and PCa.

Type of Research	Times	Research Subjects	Conclusion	References
Clinical Research	2011	932 patients treated with brachytherapy or radiation therapy.	There was an association between thickness and size of PPAT and high risk of PCa.	[14]
Clinical Research	2012	652 prostate cancer patients	The thickness of PPAT was associated with the detection rate of prostate cancer, especially high-grade PCa.	[6]
Clinical Research	2014	184 patients who underwent radical retropubic prostatectomy.	PPAT area and ratio (PPAT volume/prostate volume) were associated with high-risk PCa.	[15]
Clinical Research	2014	308 patients treated with radiotherapy.	PPAT regions were associated with PCade aggressiveness and were associated with skin color.	[5]
Clinical Research	2015	190 PCa patients undergoing MRI.	PPAT thickness was an independent predictor of PCa and high-grade PCa.	[16]
Clinical Research	2017	371 patients with PCa, 292 patients with high-grade Pca.	PPAT thickness was a potential detection metric for PCa and advanced PCa.	[17]
Clinical Research	2017	162 patients who underwent MRI prior to prostatectomy.	PPAT fat ratio correlated with PCa aggressiveness.	[18]
Clinical Research	2021	175 prostate cancer patients (mean age 62.5 years, mean prostate-specific antigen 5.4 ng/dL).	A higher periprostatic fat ratio was found to be significantly associated with a higher Gleason score by parametric magnetic resonance imaging (mpMRI).	[19]
Clinical Research	2021	175 prostate cancer patients (mean age 62.5 years, mean prostate-specific antigen 5.4 ng/dL).	Increased periprostatic fat volume was associated with disease progression in prostate cancer patients.	[19]
Clinical Research	2020	85 men with advanced PCa receiving ADT who had not received hormone therapy.	PPAT thickness was a predictor of survival in patients with advanced PCa not receiving hormonal therapy.	[20]
Basic Studies	2012	PPAT in prostate cancer patients, PC3, LNCaP.	PPAT-derived factors increased migration of PC3 and LNCaP cell lines, while PPAT had a strong proliferative effect on PC3 cell lines.	[21]
Basic Studies	2021	PPAT from 14 PCa patients (median age 62 years, median BMI 28.3) who underwent radical prostatectomy, DU145, PC3	Conditioned medium (CM) culture of PPAT promoted migration of human androgen non-dependent PCa cell lines and upregulated CTGF expression.	[22]
Basic Studies	2018	DU145, PPAT in 36 Caucasians and 36 African-Caribbeans	Fatty acid (FA) content in PPAT is associated with PCa progression.	[23]
Basic Studies	2021	PPAT in vitro culture collection of conditioned medium, DU145, PC3.	PPAT secreted IGF-1 to upregulate TUBB2B β-microtubulin heterodimer to promote resistance to doxorubicin in prostate cancer.	[24]
Basic Studies	2012	PPAT, PC-3, and LNCaP cell lines from prostate cancer patients.	PPAT increased MMP (matrix metalloproteinase) activity to regulate the microenvironment of extraprostatic tumor cells and promoted prostate cancer cell survival and migration.	[21]
Basic Studies	2019	Prostate cancer cell lines C4-2B, Du-145, and PC-3.	Free fatty acids released by PPAT promoted tumor progression by affecting the HIF1/MMP14 pathway by stimulating NOX5/ROS.	[25]
Basic Studies	2018	Primary NK cells, C4-2, 3T3-L1.	Inhibition of the IL-6/leptin-JAK/Stat3 signaling axis in adipocytes enhanced immune killing of CRPC (castration-resistant prostate cancer) cells by NK cells.	[26]
Basic Studies	2021	Adipocytes isolated from PCa patients and PC3, 22RV1	Decreased autophagic activity and increased intracellular lipid droplet content in PC3 cells after co-culture with adipocytes.	[27]
Basic Studies	2012	PPAT, LNCaP, PC3 in PCa patients.	PPAT-released pro-MMP-9 induced invasiveness of LNCaP (androgen-dependent) cells.	[28]
Basic Studies	2009	PPAT collected from patients undergoing radical prostatectomy.	PPAT regulated the aggressiveness of prostate cancer by providing IL-6.	[13]

**Table 2 cancers-14-05077-t002:** Tumorigenic effects of adipocyte-derived exosomes.

Exosome Cargo	Source	Role in Tumor	References
MMP3	3T3-L1 adipocytes	Induction of lung cancer metastasis through activation of the MMP3/MMP9 process.	[87]
miRNA-21	Cancer-associated adipocytes	Inhibition of the apoptotic process in ovarian cancer cells.	[89]
-	3T3-L1 adipocytes	Reducing degradation of caspase 3/PARP molecules in PCa and improving resistance to doxorubicin in prostate cancer.	[37]
-	Adipocytes in the obese state	Enhanced estrogen receptor expression and growth, motility and invasion, stem-cell-like properties and epithelial–mesenchymal transition of triple-negative breast cancer cells through induction of HIF-1α activity.	[90]
miR-3940-5p, miR-22-3p, miR-16-5p	Adipose mesenchymal stem cells	Inhibiting the proliferation and migration of rectal cancer.	[91]
circ-DB	3T3-L1 adipocytes	Inhibiting miR-34a and activating USP7/Cyclin A2 signaling pathway promote hepatocellular carcinoma growth and reduce DNA damage.	[92]
miR-381-3p	Adipose mesenchymal stem cells	Inhibition of apoptosis and progression of triple-negative breast cancer cells.	[93]
microRNA-1236	Adipose mesenchymal stem cells	Inhibiting SLC9A1 and Wnt/β-linked protein signaling to reduce cisplatin resistance in breast cancer cells.	[94]
-	Adipose mesenchymal stem cells	Increasing COLGALT2 expression to promote osteosarcoma proliferation and metastasis.	[95]
miR-27a-3p	3T3-L1 adipocytes	Inhibiting ICOS+ T cell proliferation and IFN-γ secretion to alter the immune microenvironment of lung adenocarcinoma.	[96]
miR-23a/b	3T3-L1 adipocytes	Targeting the VHL/HIF axis to promote HCC cell growth and migration.	[88]
hsa-miR-124-3p	Adipose mesenchymal stem cells	Inhibiting the growth and proliferation of ovarian cancer cells.	[97]
microRNA-21	Cancer-associated adipocytes	Targeting APAF1 promotes paclitaxel resistance in ovarian cancer cells.	[89]
-	Adipose mesenchymal stem cells	Mediated Wnt signaling pathway induces migration of breast cancer cells.	[98]

**Table 3 cancers-14-05077-t003:** Effects of tumor-derived exosomes on adiposity.

Exosome Cargo	Source	Role in Adipose	References
miRNA-126	Breast cancer	Decreases the uptake of glucose by fat cells and increases their secretion of lactate and pyruvate.	[99]
miRNA-155	Breast cancer	Promotes beige/brown differentiation and remodeling of adipocytes through downregulation of PPARγ expression.	[100]
ciRS-133	Gastric cancer	Regulating preadipocytes and regulating preadipocyte differentiation.	[101]
IL-6	Lung cancer	Inducing adipocyte lipolysis by mediating the STAT3 pathway.	[102]
Parathyroid hormone-related protein	Lewis lung carcinoma	Inducting lipolysis and adipose tissue browning through the PKA pathway.	[103]
miR-425-3p	A549, H1299 and AGS	Inducing white adipocyte atrophy.	[104]
miR-155	Stomach cancer cells	Inhibiting adipogenesis and promoting brown adipose differentiation via C/EPBβ pathway in adipose mesenchymal stem cells.	[105]
ciRS-133	Gastric cancer cells	Activation of PRDM16 and inhibition of miR-133 promote differentiation of preadipocytes into brown adipocytes.	[101]
circ_0004303	Gastric cancer cells	Promoting migration and invasion of adipose mesenchymal stem cells.	[106]
miR-146b-5p	Human colorectal cancer tissue	Promoting adipose tissue browning and inhibiting HOXC10 to accelerate lipolysis.	[107]
-	HepG2	Inducing adipose MSCs to differentiate into cancer-associated myofibroblasts.	[108]
Adrenomedullin	Human pancreatic cancer tissue exosomes	Activating p38 and ERK1/2 MAPK and promoting lipolysis by phosphorylating hormone-sensitive lipase.	[109]
H-ras, miR-125b, miR-155, and GTPases	C4-2B prostate cancer cells	Inducing prostate-tumor-like transformation of adipose stem cells.	[44]

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
