# Peer review of "Obesity-Related Cross-Talk between Prostate Cancer and Peripheral Fat: Potential Role of Exosomes"

_cancers, 2022, doi:10.3390/cancers14205077_

Round 1

Reviewer 1 Report

Review (Comments to the Author)

For the paper / review – No. cancers-1939470 - entitled: "Obesity-Related cross-talk

between Prostate Cancer and Peripheral Fat: Potential Role of Exosomes" are minor and

major aspects that should be considered.

The theme of the paper about the feedback mechanisms between prostate cancer and

periglandular adipose and the role of exosomes as mediators of signal exchange to

understand obesity as a risk factor for prostate cancer is very interesting. The review

provides compiled important aspects for future prostate cancer management.

Table 1:

• I would recommend supplementing the respective year of the publications.

• The abbreviations should be explained as required in MDPI papers.

Introduction:

• The method ‚Nuclear magnetic resonance spectroscopy‘ should be explained.

• The authors should rephrase the sentence "NMR spectroscopy showed that Nuclear

magnetic resonance spectroscopy shows that..." on page 5.

• Is reference 65 the correct reference for NMR? Please check and change if

necessary.

• The authors used the terms ‚microspectroscopy‘, ‚nuclear magnetic resonance

spectroscopy‘ and ‚magnetic resonance spectroscopy‘. Please specify the different

terms for the reader.

Conclusion:

• The authors should check the sentences, especially the text on the exosomes, as

some passages appear twice here.

• All definitive statements in conclusions should be supported with references.

Editorial note:

• No blanks in Funding, Competing interests, Acknowledgments and Authors‘

contributions.

• Note upper and lower case in the chapter 2. „Cross-talk between PCa and PPAT“ on

page 3.

• All abbreviations should be explained. For example in chapter 2.1. „PPAT promotes

….“ – last paragraph: abbreviation „MMP“ or for example „IL-8“ or for example „CGI-

58“ and so on.

• In the discussion ‚prostate cancer‘ should be abbreviated.

• First paragraph in the discussion on page 13: „the“ is used twice.

Author Response

Dear Reviewer 1,

We are very grateful to you for your review of our manuscript, which has provided us with positive comments and constructive suggestions. Accordingly, we have revised the manuscript in line with your suggestions. The following is a response to the specific comments:

Table 1:

  • I would recommend supplementing the respective year of the publications.

Response: We have added the respective year of publication.

  • The abbreviations should be explained as required in MDPI papers.

 Response: We have abbreviated as required for MDPI papers.

Introduction:

  • The method ‚Nuclear magnetic resonance spectroscopy‘ should be explained.

Response: We have explained the method "Nuclear magnetic resonance spectroscopy".

  • The authors should rephrase the sentence "NMR spectroscopy showed that Nuclear magnetic resonance spectroscopy shows that..." on page 5.

Response: We have rewritten the sentence on page 5 “NMR spectra show NMR spectra show ......” See the revised version.

  • Is reference 65 the correct reference for NMR? Please check and change if

necessary.

Response: We have revised reference 65.

  • The authors used the terms ‚ ‘microspectroscopy, ‚nuclear magnetic resonance. Spectroscopy and ‚magnetic resonance spectroscopy’. Please specify the different

terms for the reader.

Response: We have described the above terms in detail. See the revised version

Conclusion:

  • The authors should check the sentences, especially the text on the exosomes, as some passages appear twice here.

Response: We have checked the sentences and made changes to the sentences. Removed some repetitive sentences. See the revised version.

  • All definitive statements in conclusions should be supported with references.

Response: We have checked the sentences and added references to efinitive statements.

Editorial note:

  • No blanks in Funding, Competing interests, Acknowledgments and Authors‘contributions.

Response: We have made revisions based on the comments.

  • Note upper and lower case in the chapter 2. „Cross-talk between PCa and PPAT“ on page 3.

Response: We have made revisions based on the comments.

  • All abbreviations should be explained. For example in chapter 2.1. „PPAT promotes….“ – last paragraph: abbreviation „MMP“ or for example „IL-8“ or for example „CGI-58“ and so on.

Response: We have made revisions based on the comments.

  • In the discussion ‚prostate cancer‘ should be abbreviated.

Response: We have made revisions based on the comments.

  • First paragraph in the discussion on page 13: „the“ is used twice.

Response: We have made revisions based on the comments.

Thanks again for your positive comments and encouragement.

Yours Sincerely,

Shangzhi Feng

Reviewer 2 Report

In this review paper review the role played by obesity in the development and progression of prostate cancer. They discuss the molecular signals that allow for the bidirectional communication between prostate cancer and periprostatic adipose tissue. This includes a detailed review of the literature concerning the rile played cy signals carried by exosomes in this capacity. This is an important topic, and the review is able to successfully discuss the available literature and provide insights into the results of previous studies and the future direction and benefits of these studies. I would like to congratulate the authors on a well written and thought-out manuscript. There are however a few concerns with the language and images which are discussed below. I recommend that the review is accepted following minor corrections.

General concerns.

Throughout the manuscript and is used incorrectly when it comes to lists. All items in the list should be separated by a comma and the fila item should be preceded by “and”.

In vivo and in vivo are bot italicized throughout the paper. There are some instabces where they are, but many where they are not.

Acronyms are not always defined the first time they are used.

Figure legends are not adequate to explain the images.

In the abstract

and the prospect of using adipose tissue exosomes- Use the exosomes for what. Authors must be more specific with certain statements.

In the introduction

And high mortality and biochemical recurrence- This is an instance of the incorrect placement of and. Also what do the authors mean by biochemical recurrence.

In section 2

Epidemiological evidence related to the effect of obesity on PCa progression, the association between PPAT

thickness and PCa aggressiveness and the high expression of inflammatory factors in PPAT compared with normal glands has suggested that changes in the local adipose tissue of the prostate body can influence the behavior of PCa.

This sentence is too long and clumsy and should be re-written.

preadipocytes triggered by PCa cell culture medium undergo tumor-like transformations, Including genetic variability, EMT, and tumor-like lesion formation

Including should not be capitalised and EMT should be defined the first time it is used.

and inflammatory cytokines, which play an important role in paracrine and endocrine influencing local

and systemic metabolic responses.

This is very unclear paracrine and endocrine what?

For example, PCa-conditioned medium (CM) significantly affects preadipocytes than non-malignant PCa cells.

The word more is missing (more than)

In section 3.

Binding inhibits apoptosis and promotes chemoresistance in cancer cells 90.

Binding to what?

The adipocyte-secreted exosome miR-27a-3p targets ICOS,

Exosome containing this miRNA?

Furthermore, also as mentioned be-fore, PCa cells do influence the status of peripheral adiposity

Also must be deleted

In section 4

Alix, four-span proteins,

Is four-span the correct term?

Obesity also alters the gene expression profile of PPAT, which increases cancer cells and decreases immune surveillance,

Increases cancer cells? Be more specific increases cancer cells what?

EVs also causes AKT activation

Cause instead of causes

Depots should be deposits

In section 5

accelerates the further spread of cancer in certain potential ways. Especially for the peripheral tissues of cancerous tissues, this

Certain potential ways rather use the word possible

In section 6

In furthermore, some ideas may

Delete In and capitalise furthermore

multiple recurrences of the cancer as well as the maintaining high carcinogenicity and carcinogenicity in the body and thus affecting the use of chemotherapeutic drugs etc, which brings thoughts for radical surgery for PCa with simultaneous resection

Remove the

Brings thoughts – this is too informal and clumsy sounding. – leads to the adoption of radical surgery

their exact value in cancer.

In cancer what?

on the the bioinformatic molecules

The is repeated

Bioinformatic- generally refers to computational tools or techniques

tumorigenic controversy,

I am not sure what the authors are referring to here

significantly reduce melanin levels than placebo

Placebo treatments?

become a major concern, and adipose tissue, as the most important source of

what does and adipose tissue refer to here?

Author Response

Dear Reviewer 2,

We are very grateful to you for your review of our manuscript, which has provided us with positive comments and constructive suggestions. Accordingly, we have revised the manuscript in line with your suggestions. The following is a response to the specific comments:

General concerns.

Throughout the manuscript and is used incorrectly when it comes to lists. All items in the list should be separated by a comma and the fila item should be preceded by “and”.

Response: We have made revisions based on the comments. However, we did not put the word "and" in front of the fila item because we could not understand the "fila item".

In vivo and in vivo are bot italicized throughout the paper. There are some instabces where they are, but many where they are not.

Response: We have made revisions based on the comments. See the revised version.

Acronyms are not always defined the first time they are used.

Response: We have made revisions based on the comments. See the revised version.

Figure legends are not adequate to explain the images.

Response: We have added the necessary sentences to explain the image. See the revised version.

In the abstract

and the prospect of using adipose tissue exosomes- Use the exosomes for what. Authors must be more specific with certain statements.

Response: We have made revisions based on the comments. See the revised version.

In the introduction

And high mortality and biochemical recurrence- This is an instance of the incorrect placement of and. Also what do the authors mean by biochemical recurrence.

Response: We have made revisions based on the comments. See the revised version.

In section 2

Epidemiological evidence related to the effect of obesity on PCa progression, the association between PPAT thickness and PCa aggressiveness and the high expression of inflammatory factors in PPAT compared with normal glands has suggested that changes in the local adipose tissue of the prostate body can influence the behavior of PCa.

This sentence is too long and clumsy and should be re-written.

Response: We have made revisions based on the comments. See the revised version.

preadipocytes triggered by PCa cell culture medium undergo tumor-like transformations, Including genetic variability, EMT, and tumor-like lesion formation

Including should not be capitalised and EMT should be defined the first time it is used.

Response: We have made revisions based on the comments. See the revised version.

and inflammatory cytokines, which play an important role in paracrine and endocrine influencing local and systemic metabolic responses.

This is very unclear paracrine and endocrine what?

Response: Paracrine secretion is a mode of secretion in which cellular secretions do not enter the circulation, but act by diffusion on neighboring target cells for intercellular signaling, producing mainly local effects. Endocrine: A form of secretion in which the hormone produced by the secretory cells enters directly into the body fluids to produce effects on target cells mediated by the body fluids, producing mainly systemic responses.

For example, PCa-conditioned medium (CM) significantly affects preadipocytes than non-malignant PCa cells.

The word more is missing (more than)

Response: We have made revisions based on the comments. See the revised version.

In section 3.

Binding inhibits apoptosis and promotes chemoresistance in cancer cells 90.

Binding to what?

Response: We have made revisions based on the comments. See the revised version. 

The adipocyte-secreted exosome miR-27a-3p targets ICOS,

Exosome containing this miRNA?

Response: We have made revisions based on the comments. See the revised version.

Furthermore, also as mentioned be-fore, PCa cells do influence the status of peripheral adiposity

Also must be deleted

Response: We have made revisions based on the comments. See the revised version.

In section 4

Alix, four-span proteins,

Is four-span the correct term?

Response: We have removed the four-span proteins as they already contain CD81 and CD36. The four-span proteins are the transmembrane 4 superfamily. See the revised version.

Obesity also alters the gene expression profile of PPAT, which increases cancer cells and decreases immune surveillance,

Increases cancer cells? Be more specific increases cancer cells what?

Response: We have made revisions based on the comments. See the revised version.

EVs also causes AKT activation

Cause instead of causes

Response: We have made revisions based on the comments. See the revised version.

Depots should be deposits

Response: We have made revisions based on the comments. See the revised version.

In section 5

accelerates the further spread of cancer in certain potential ways. Especially for the peripheral tissues of cancerous tissues, this

Certain potential ways rather use the word possible

Response: We have made revisions based on the comments. See the revised version.

In section 6

In furthermore, some ideas may

Delete In and capitalise furthermore

Response: We have made revisions based on the comments. See the revised version.

multiple recurrences of the cancer as well as the maintaining high carcinogenicity and carcinogenicity in the body and thus affecting the use of chemotherapeutic drugs etc, which brings thoughts for radical surgery for PCa with simultaneous resection

Remove the

Brings thoughts – this is too informal and clumsy sounding. – leads to the adoption of radical surgery

Response: We have made revisions based on the comments. See the revised version.

their exact value in cancer.

In cancer what?

Response: We have made revisions based on the comments. See the revised version.

on the the bioinformatic molecules

The is repeated

Bioinformatic- generally refers to computational tools or techniques

Response: We have made revisions based on the comments. See the revised version.

tumorigenic controversy,

I am not sure what the authors are referring to here

Response: Mesenchymal stem cells are potentially carcinogenic. Mesenchymal stem cells are potentially carcinogenic. See also "Hong IS, Lee HY, Kang KS. Mesenchymal stem cells and cancer: friends or enemies? Mutat Res. 2014 Oct;768:98-106. doi: 10.1016/j.mrfmmm.2014.01.006. Epub 2014 Feb 7. PMID: 24512984."

significantly reduce melanin levels than placebo

Placebo treatments?

Response: In the field of medical research, the use of placebos often means that people involved in clinical trials are not made aware of what treatment they are receiving. To determine whether a drug in a study is truly effective, researchers judge whether patients who take the drug in a clinical trial show better results than those who take a placebo.

become a major concern, and adipose tissue, as the most important source of

what does and adipose tissue refer to here?

Response: The term "adipose tissue" here refers to human adipose tissue, which is often used for the extraction of human-derived adipose mesenchymal stem cells.

Yours Sincerely,

Shangzhi Feng

Reviewer 3 Report

The manuscript reviews the main findings about the crosstalk between PCa and PPAT. Overall, the manuscript is clear and well-written. The potential role of EVs is well discussed, and figures and tables are interesting too. I do not have any concern about this article and I recommend it for publication in the present form.

Author Response

Dear Reviewer 3,

We are very grateful for your review of our manuscript and are honoured to have your permission to publish this manuscript. We would be grateful if you could review our revised manuscript again.

Yours Sincerely,

Shangzhi Feng

Round 2

Reviewer 1 Report

No further comments.